# A Multiple-Input Multiple-Output Reservoir Computing System Subject to Optoelectronic Feedbacks and Mutual Coupling

**DOI:** 10.3390/e22020231

**Published:** 2020-02-18

**Authors:** Xiurong Bao, Qingchun Zhao, Hongxi Yin

**Affiliations:** 1School of Information and Communication Engineering, Dalian University of Technology, Dalian 116024, China; bxr@imnu.edu.cn; 2College of Physics and Electronic Information, Inner Mongolia Normal University, Hohhot 010022, China; 3School of Computer and Communication Engineering, Northeastern University at Qinhuangdao, Qinhuangdao 066004, China; zhaoqingchun@neuq.edu.cn

**Keywords:** reservoir computing, multiple-input multiple-output, optoelectronic feedback, mutual coupling, optical packet header recognition, digital speech recognition

## Abstract

In this paper, a multiple-input multiple-output reservoir computing (RC) system is proposed, which is composed of multiple nonlinear nodes (Mach–Zehnder modulators) and multiple mutual-coupling loops of optoelectronic delay lines. Each input signal is added into every mutual-coupling loop to implement the simultaneous recognition of multiple route signals, which results in the signal processing speed improving and the number of routes increasing. As an example, the four-route input and four-route output RC is simultaneously realized by numerical simulations. The results show that this type of RC system can successfully recognize the four-route optical packet headers with 3-bit, 8-bit, 16-bit, and 32-bit, and four-route independent digital speeches. When the white noise is added to the signals such that the signal-to-noise ratio (SNR) of the optical packet headers and the digital speeches are 35 dB and 20 dB respectively, the normalized root mean square errors (NRMSEs) of the signal recognition are all close to 0.1. The word error rates (WERs) of the optical packet header recognition are 0%. The WER of the digital speech recognition is 1.6%. The eight-route input and eight-route output RC is also numerically simulated. The recognition of the eight-route 3-bit optical packet headers is implemented. The parallel processing of multiple-route signals and the high recognition accuracy are implemented by this proposed system.

## 1. Introduction

The reservoir computing (RC) based on the delay structure is a simple and efficient machine learning algorithm that is suitable for processing the real-time signal. The higher dimensional computation for the emerging artificial intelligence can be realized due to RC with the simple hardware structure and the short-time training algorithm. Compared with the approach of realizing artificial neural networks on a personal computer by a software, the RC can be implemented by hardware and can improve the performance of traditional artificial neural networks [1,2,3,4,5,6,7].

The hardware realization of the RC based on optical devices has many advantages such as higher bandwidth, lower power consumption, parallel computation, and so forth. It can process signals quickly and efficiently; as a result, it has attracted extensive attention. Paquot et al. proposed an optoelectronic RC system composed of a Mach–Zehnder modulator subject to optoelectronic feedback, which was successfully applied to signal recognition and prediction [8]. The channel equalization and digital speech recognition were achieved by an all-optical RC system employing a semiconductor optical amplifier as a nonlinear node [9]. A RC structure consisting of a semiconductor laser subject to optical feedback was proposed by Hicke et al [10]. Nguimdo presented a RC system including a semiconductor ring laser with co-directional optical feedback or cross-directional optical feedback [11]. Another optoelectronic RC setup with an imbalanced birefringent interferometer to form multiple feedback delay lines was proposed in reference [12]. A RC system consisting of a phase modulator and additional multiple optical feedbacks was designed by Akrout et al [13]. All of the above-mentioned reservoirs are composed of a single nonlinear node subject to additional feedback loops, which can only process a single-route input signal. In [8,9,10,11,12], the signal processing results were all obtained in an ideal situation without any noise. Then, the further research on the RC with delay structure had been carried out, including the study on the simultaneous processing of more routes of signals to extend the signal processing capacity. Brunner et al. designed a RC setup using a feedback loop including a semiconductor laser, which realized the simultaneous high-speed processing of two input signals through injecting input signals to the feedback loop and the laser from its driving current [14]. Nguimdo brought forward a semiconductor ring laser employed as a nonlinear node subject to two optical feedbacks; each signal was injected into every feedback loop, which resulted in two-route signals being simultaneously processed [15,16]. A vertical-cavity surface-emitting laser (VCSEL) subject to double optical feedback and optical injection is utilized as a nonlinear node. The parallel information processing of this RC system is implemented based on the dynamical responses of the X polarization component and Y polarization component in the VCSEL [17]. Some parallel RC systems with multiple feedback loops attached to a single nonlinear node are also designed, which can process two or three route independent signals simultaneously [18,19]. An all-optical reservoir composed of a semiconductor laser with two optical feedbacks was proposed that successfully realized the simultaneous recognition of three route signals [18]. The simultaneous recognition of two routes of independent optical packet headers is implemented by an all-optical RC utilizing a semiconductor ring laser with double optical feedbacks [19]. However, the RC systems proposed above can only process at most three route input signals concurrently. The anti-noise capabilities have not been analyzed in detail.

In this paper, a multiple-input multiple-output optoelectronic RC scheme with Mach–Zehnder modulators is proposed, which is constituted of multiple nonlinear nodes and multiple mutual coupling loops of optoelectronic delay lines. The Mach–Zehnder modulators act as the nonlinear nodes, and each input signal to be identified is injected into each feedback loop, which can achieve the parallel processing of the multiple-route input signals. The numerical simulations of the four-input four-output setup are performed where four-route optical packet headers with four lengths—3-bit, 8-bit, 16-bit, and 32-bit—are recognized simultaneously. It is equivalent to an optical switch of switching nodes in the hybrid optical/electrical packet switching networks. Moreover, four-route digital speech signals are recognized simultaneously. The influence of the principle parameters, including the feedback strength and the noise, on the signal recognition results is analyzed. We also present the numerical simulation results of the eight-route input and eight-route output RC setup. The recognition of the 8-route 3-bit optical packet headers is implemented. It is proved that the system has a certain ability to resist noise interference.

## 2. Numerical Simulation Model

The reservoir computing process is explained in detail by taking the single-input RC as an example. The schematic diagram is shown in Figure 1a, which is composed of an input layer, a reservoir layer, and an output layer. The reservoir as the intermediate layer consists of a feedback loop containing many virtual nodes, which conducts as a delayed nonlinear dynamic system. The virtual nodes on the delay line in the dynamic system replace the neurons in a traditional recurrent neural network. These virtual nodes will generate lots of transient states for subsequent information processing. Concerning the input layer, the input signal is real-time sampled to the discrete signal where the sampling rate is 1/τ, and τ is the feedback delay time. In order to process the signal successfully, the reservoir is required to generate a large number of different nonlinear transient states. Hence, each discrete value of the input needs to be multiplied by the pre-processing mask with the sampling interval τ before being entered serially into the reservoir. The mask is a binary random matrix consisting of 0’ and 1’ values. Through the pre-processing of the input signal, the computing period τ of each discrete value will be divided into *M* time intervals. Therefore, the reservoir generates *M* different nonlinear transient states *x*_r_(t) (r = 1, 2,..., *M*). Finally, the reservoir states *x*_r_(t) are acquired at the sampling rate of *τ*/*M* at the output layer. After a linear sum of the reservoir states is taken, the output value is obtained [20,21,22,23]. These coefficients in this linear sum are called the output weights, which are the only ones that need be determined by training in the system. A simple linear regression algorithm can be employed to minimize the difference between the actual computing output value and the desired output value [24]. Their difference is represented by the normalized root mean square error (NRMSE) and the word error rate (WER), as described in Equations (1) and (2). The smaller the values of NRMSE and WER, the closer the actual output value is to the desired output value.
(1)$$NRMSE=\sqrt{\frac{\langle \left(\hat{y}(t\right)-y(t){)}^{2}\rangle }{\langle \left(y(t\right){-\langle y{\left(t\right)\rangle }_{t}{)}^{2}\rangle }_{t}}}=\sqrt{\frac{\frac{1}{n}\sum_{t=1}^{n}{\left(\hat{y}(t)-y(t)\right)}^{2}}{\frac{1}{n}\sum_{t=1}^{n}{\left(y(t)-\frac{1}{n}\sum_{t=1}^{n}y(t)\right)}^{2}}}$$
(2)$$WER=\frac{N_{all}-N_{correct}}{N_{all}}\times 100\%$$
where *ŷ(t)* is the actual output value, *y(t)* is the desired output value, *n* is the vector length of the output signal, *N_all_* is the number of all the input signals, and *N_correct_* is the number of signals correctly recognized. In our numerical simulations, 10 training repetitions will be carried out with different arrays, and the average values of the results of the 10 repetitions will be taken as their final values.

A multiple-input multiple-output parallel RC system is proposed, as shown in Figure 1b, where *N* nonlinear nodes (Mach–Zehnder modulators, MZMs) and *N* mutual coupling loops of the optoelectronic feedbacks are utilized to form a reservoir. At the reservoir layer, a semiconductor laser diode (LD) outputs a coherent light, and then it is amplified by an erbium-doped fiber amplifier (EDFA). Then, the light is divided into *N* routes through an optical coupler (OC), and afterwards, it is input into *N* MZMs. *N*-route optical signals from *N* MZMs combine together into a one-route electric signal after the optoelectronic delays; then, it is divided into *N*-route signals through an electric power divider (D_N + 1_). Each input signal to be identified, input*_i_* (*i* = 1, 2, 3, ……, *N*), is added to every route loop, and together they are fed back to the radio frequency (RF) terminal of each MZM. Consequently, *N*-route input signals can be recognized simultaneously. The optoelectronic delays are composed of the optical delay lines (ODL*_i_*, *i* = 1, 2, 3, ……, *N*), photodetectors (PD*_i_*, *i* = 1, 2, 3, ……, *N*), band-pass filters (BPF*_i_*, *i* = 1, 2, 3, ……, *N*), electric power amplifiers (AMP*_i_*, *i* = 1, 2, 3, ……, *N*), and electric power distributers (D*_i_*, *i* = 1, 2, 3, ……, *N*). ODL*_i_* is used for the time delay. PD*_i_* converts an optical signal into an electrical signal. BPF*_i_* is employed for band limitation. AMP*_i_* amplifies an electrical signal. D_i_ is used for collecting the reservoir states.

The dynamic model of the proposed RC setup can be expressed by the delay differential Equations (3) and (4), where the subscript of variables represents *N* loops. There are actually 2*N* equations representing the system model [25,26].
(3)$$\tau_{Li}\frac{dx_{i}(t)}{dt}=-(1+\frac{\tau_{Li}}{\tau_{Hi}})x_{i}(t)-y_{i}(t)+\beta_{i}\cos^{2}[\frac{1}{N}\sum_{j=1}^{N}x_{j}(t-\tau_{j})+\phi_{i}+\gamma_{i}J_{i}(t)+n_{i}(t)]$$
(4)$$\tau_{Hi}\frac{dy_{i}(t)}{dt}=x_{i}(t)\begin{array}{cccc} & & & i=1,2,3,\cdots \cdots ,N \end{array},$$
where *x*_i_(*t*) = π*v_i_*(*t*)/2*v*_πRF_ is the normalized bias voltage of MZM, *v_i_*(t) is the voltage of each loop, and *v*_πRF_ is the input half-wave voltage of MZM. In order to facilitate the numerical solution, the integral of *x_i_*(*t*) is set as y*_i_*(*t*), as shown in Equation (4). *ϕ_i_* is the phase offset, which is determined by the direct current (DC) offset voltage of the MZM. *f_Hi_* and *f_Li_* are the cutoff frequencies for the high frequency and the low frequency, respectively, *τ_Hi_* = 1/2π*f_Hi_*, *τ_Li_* = 1/2π*f_Li_*. *β_i_* is the feedback strength. *τ_j_*(*t*) (*j* = 1, 2, 3, ⋅⋅⋅, *N*) is the feedback delay time determined by the length of ODL*_i_*. *γ_i_* (*i* = 1, 2, 3, ⋅⋅⋅, *N*) is the gain coefficient of the input signal. *J_i_*(t) (*i* = 1, 2, 3, ⋅⋅⋅, *N*) is the serial input signal of the reservoir. *n_i_*(*t*) is the white noise added to the signal. The values of the system parameters are shown in Table 1. The main noise of the whole setup includes the noise of the input signals, the insertion loss of the MZMs, and the thermal noise of photodetectors. For the performance analysis of the RC system, the influence of the noise of the input signals is mainly analyzed. The noise of the MZMs and photodetectors is ignored for the sake of simplification here.

## 3. Simulation Results of Signal Recognitions

### 3.1. Four-Route Optical Packet Header Recognition

Concerning an optical packet switching network, a correct recognition of the optical packet header is crucial, which can ensure that the data are transmitted to a correct route [27,28]. If the multiple-input multiple-output optoelectronic RC system can successfully realize the simultaneous recognition of the multiple-route optical packet headers, it can be employed as an important equipment for the optical network switching nodes. For our numerical simulations, we test the signal recognition of RC with eight routes or even more routes. Due to too much data for a RC simulation system with various input numbers and the limited length of the paper, in Section 3.1 and Section 3.3, we only take *N* = 4 and *N* = 8, namely, the 4-input 4-output and 8-input 8-output RC systems based on the optoelectronic feedbacks and mutual coupling as examples, to realize the simultaneous recognition of 4-route and 8-route 3-bit (000-111) with eight types of the optical packet headers, respectively. Firstly, each bit of the optical packet header is sampled at six points, and then the white noise is added with a signal-to-noise ratio (SNR) of 20 dB. This operation is repeated 10 times for each type of the optical packet header. Therefore, there are a total of 80 optical packet headers in the database. These optical packet headers are multiplied by the pre-processing mask; then, they are entered serially into the four-route input terminals of the reservoir. The number of virtual nodes in the reservoir is set to 400. In order to facilitate the feedback strengths of the reservoir, the feedback strengths of the four-route loops are set to be the same in the numerical simulations. At the output layer, the reservoir states are collected for training, where 80% of the reservoir states are used for training and 20% are used for testing.

The impact of the feedback strength and the noise on the optical packet header recognition results is analyzed. The feedback strength is an important parameter that affects the dynamical characteristics of the reservoir, which directly influences the signal recognition results. The influence of the feedback strength on the optical packet header recognition is shown in Figure 2. The recognition results of a one-route signal are only given in the following figures and tables, because they are similar to the recognition results of the other three-route signals. As shown in Figure 2, when the feedback strengths are 1.5 GHz, the reservoir is in the critical state between the stable state and chaos. The training NRMSE and testing NRMSE for the optical packet header recognition are 0.0570 and 0.0730, respectively. The WER is 0%. In this case, the signal recognition errors are the smallest. When the feedback strength is less than 1.5 GHz, the reservoir is in a stable state with monotonous values. Hence, the signal identification errors are larger. When the feedback strength is greater than 1.5 GHz, the reservoir gradually enters a chaotic state. At this time, although the training NRMSE and WER show little change, the testing NRMSE gradually gets larger. The training and the testing NRMSE are no longer close to each other. The output coefficients trained are no longer suitable for the testing data. 

The results of the 3-bit optical packet header recognition are shown in Figure 3. The horizontal axis of each sub-graph in Figure 3 is the number of sampling points of the 3-bit optical packet headers with a length of 18. The vertical axis is the category of the optical packet headers. When the output value of the RC is closer to ‘+1’, it will be shown in red, which means that the recognition results of the optical packet header are more accurate. When the output value is close to ‘−1’, it will be shown in blue. Figure 3a presents the desired output of the 3-bit optical packet headers, while Figure 3b shows the actual recognition output. The closer both are, the better the signal recognition effect will be. As can be seen from Figure 3, the recognition results for the four-route of 3-bit optical packet headers are good. The above results are obtained with the SNR of the input signal with 20 dB. For practical applications, the SNR is generally within the range between 20 and 40 dB, indicating that the system has certain anti-noise capability. 

Figure 4 shows the impact of the noise added into the input signal on the optical packet header recognition results. It confirms that the signal recognition accuracy increases with the decrease of noise. Meanwhile, it reveals that when the SNR of the input signal is greater than 20 dB, the NRMSEs of the recognition results are all less than 0.1, and the WERs are all 0%. Therefore, the signal recognition results are good. However, when the SNR of the input is less than 20 dB, the testing NRMSE of the optical packet header recognition is much larger than 0.1, and the WER is larger than 0%. The simulation results show that the proposed setup meets the practical applications.

The bit number of the optical packet headers is expanded to approach the actual application. The simultaneous recognition of the four-route 8-bit, 16-bit, and 32-bit optical packet headers is realized by simulation, respectively. Eight types of 8-bit optical packet headers for recognizing are randomly 66H, 22H, 44H, AAH, 82H, 32H, 74H, and E4H, where the symbol “H” at the end of these data represents the hexadecimal number. Eight kinds of 16-bit optical packet headers are randomly taken as 0097H, 219CH, 5262H, 64CFH, 98F0H, B98CH, CC35H, and FD39H. Eight kinds of 32-bit optical packet headers are randomly selected as CA764206H, A66F4482H, 0A0A0015H, CA744242H, 8B365204H, A33B4492H, 2A0A2035H, and 86754252H. In the pre-processing of the input signal, every bit of 8-bit and 16-bit optical packet header is sampled at four points. Every bit of 32-bit optical packet header is sampled at two points. Then, white noise is added. After multiplying with the mask, they are respectively serially input to the four-route input terminals of the reservoir. The parameters for the reservoir remain the same. The reservoir states are collected at the output layer. The signal recognition results are obtained by training and testing. The recognition results for the optical packet headers with different lengths are shown in Table 2. In this table, the SNR of the input signal is 20 dB and 35 dB separately. When the SNRs are 20 dB and 35 dB separately, the recognition results of the four routes of the 8-bit optical packet headers are better. The NRMSEs are all less than 0.1, and the WERs are all 0%. For the recognition of the four routes of the 16-bit and 32-bit optical packet headers, the NRMSEs are greater than 0.1 when the SNR is 20 dB, and the WERs are all 0%. When the SNR increases to 35 dB, the recognition NRMSEs for the 16-bit and 32-bit optical packet headers are all close to 0.1, the WERs are all 0%, which means that the signal recognition accuracy is higher.

### 3.2. Digital Speech Recognition

Similarly, here we take *N* = 4 again, i.e., the four-input and four-output RC system based on the optoelectronic feedbacks and mutual coupling as an example to implement the four-route digital speech recognition. The digital speech recognition is a typical standard task to verify the performance of RC. The digital speech data are taken from the TIMIT (Texas Instruments and Massachusetts Institute of Technology) corpus [26,29]. The digital speech database contains a total of 500 speeches from five people who speak the figure 0 to 9 a total of 10 times. First, the signal characteristics of each digital speech are extracted by the Lyon model; then, they are divided into 86 channel data so that each speech signal becomes an 86×S two-dimensional signal, where S is the time-domain data length of each channel. The pre-processing mask is an *N*×86 two-dimensional matrix with a sparse value of ‘1’. The digital speech needs to be multiplied by the mask to be transformed an *S*×*N* matrix input signal, and then, it is transmitted to each route of the input layer respectively after adding the noise. The number of the virtual nodes on each feedback loop for the reservoir is 400. Since more different states of the reservoir are required, the division interval of the feedback delay time becomes shorter, with a value of 0.1 ns. Other parameters remain unchanged. At the output layer, the reservoir states are collected, and the output weights are trained. Most (80%) of data are used for training, and 20% are used for testing. The four routes of the digital speech recognition results are similar. The following table and figures show one of the recognition results of the four-route digital speeches. It can be seen from Table 3 that the NRMSEs for training and testing are around 0.1, and the WER is 1.4% without input noise. Considering the actual situation, the digital speech recognition results are analyzed when the SNR is between 20 and 40 dB. When the noise is added into the signal such that the input SNR is 20 dB and 30 dB separately, the NRMSEs are close to 0.1, and the WER are all 1.6%. Hence, the digital speech recognition results are good. When the SNR is 10 dB, the training and testing NRMSEs of signal recognition are no longer similar. The testing NRMSE is larger; the WER reaches 14.6%. Therefore, the speech signal recognition errors are larger. Figure 5 shows the results of speech recognition with the SNR of 20 dB. The horizontal axis of each sub-graph is the length of data in each channel of speech signal, while the vertical axis indicates the 10 categories of digital speech, respectively. It can be seen that the system can basically realize the simultaneous recognition of four-route digital speech signals with noise.

### 3.3. Eight-Input Eight-Output Optoelectronic RC

When N = 8, the eight-input eight-output optoelectronic RC system with eight nonlinear nodes and eight optoelectronic feedback loops is established. Eight routes of 3-bit (000~111) optical packet headers recognition are realized utilizing this system. The internal parameters of the eight-route reservoir are the same as those described in Section 3.1, and the white noise with a SNR of 20 dB is added into this RC system. Similarly, 80% of the data is utilized for training, while the remaining 20% is employed for testing. For simplification, one of the eight routes of the 3-bit optical packet header recognition is demonstrated, as shown in Figure 6. The training and testing NRMSEs are 0.0566 and 0.0890 respectively, while the WER is 0%. Figure 6a,b show the desired output and the actual output of the optical packet header recognition, respectively. It can be seen that both are very close. Compared with the 3-bit optical packet header recognition achieved by the four-input four-output optoelectronic RC in Section 3.1, the testing NRMSE has increased a little for the same numerical simulation parameters. With the increase of the bit number of the optical packet header, the signal recognition error utilizing the eight-input eight-output optoelectronic RC system will increase.

## 4. Conclusions

This paper proposes an *N*-input and *N*-output RC scheme based on the optoelectronic feedback and mutual coupling, which consists of *N* Mach–Zehnder modulators and *N* optoelectronic feedback loops. This setup can simultaneously recognize the *N* parallel input signals. In the above-mentioned numerical simulations, firstly taking *N*=4 as an example for simplicity, the recognition of the four-route optical packet headers for different lengths and the recognition of the four-route digital speech signals are carried out. The eight-route input and eight-route output RC is also numerically simulated. The recognition of the eight-route 3-bit optical packet headers is realized. Thus, the multiple-input multiple-output RC system based on the optoelectronic feedbacks and mutual coupling proposed in this paper not only increases the route number of the input signals for simultaneous recognizing, but also has a certain anti-noise interference capability.

## Figures and Tables

**Figure 1 entropy-22-00231-f001:**
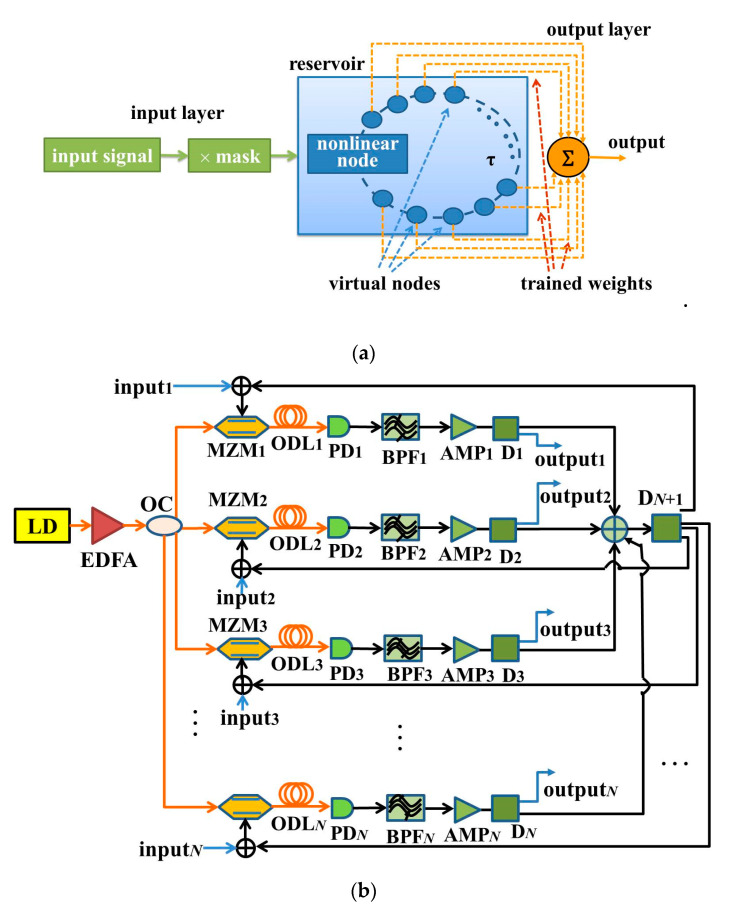
(**a**) Schematic of the RC (reservoir computing) based on a single nonlinear node subject to delay feedback; (**b**) Multiple-input multiple-output RC based on optoelectronic feedbacks and mutual coupling. LD: laser diode, EDFA: erbium-doped fiber amplifier, OC: optical coupler, MZM: Mach–Zehnder modulator, ODL: fiber-optic delay line, PD: photodetector, BPF: band-pass filter, AMP: electric amplifier, D: electric power divider.

**Figure 2 entropy-22-00231-f002:**
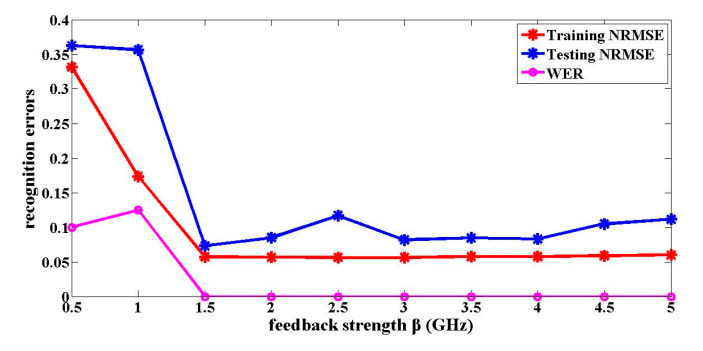
Relationship between the optical packet header recognition errors and the feedback strength β.

**Figure 3 entropy-22-00231-f003:**
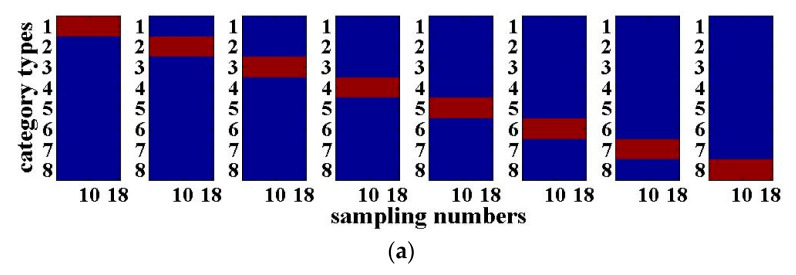
4-input 4-output RC recognition results of the 3-bit optical packet headers for the signal-to-noise ratio (SNR) of 20 dB: (**a**) Desired output; (**b**) Actual output.

**Figure 4 entropy-22-00231-f004:**
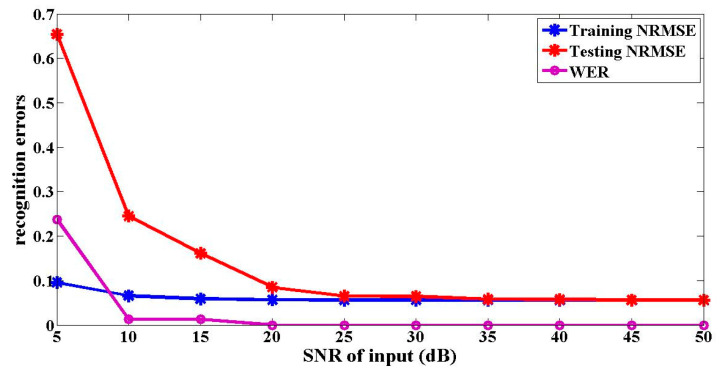
Relationships between the optical packet header recognition errors and the different SNRs of the inputs.

**Figure 5 entropy-22-00231-f005:**
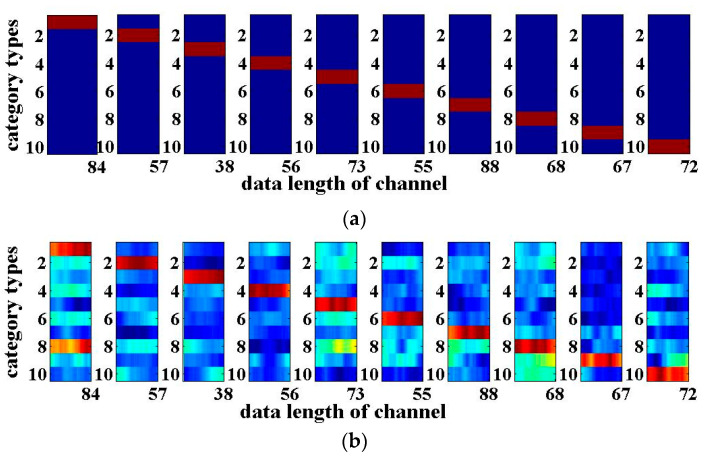
Digital speech recognition results for SNR of 20 dB: (**a**) Desired output; (**b**) Actual output.

**Figure 6 entropy-22-00231-f006:**
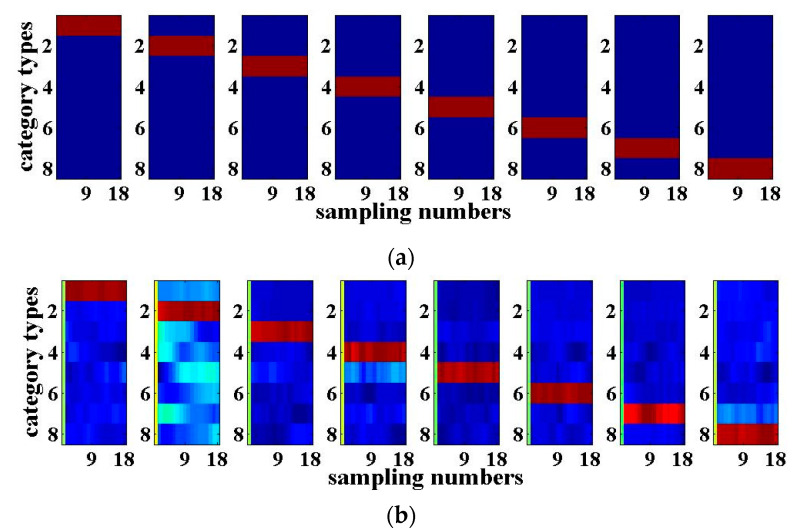
8-input 8-output RC recognition results of 3-bit optical packet headers when the SNR is 20 dB. (**a**) Desired output; (**b**) Actual output.

**Table 1 entropy-22-00231-t001:** Parameter values of the reservoir computing (RC) (*i* = 1, 2, 3, ⋅⋅⋅, N).

Symbol	Parameter	Value
*ϕ_i_*	offset phase of the MZM	–π/4
*τ_Hi_*	high-frequency cutoff characteristic time	19.89 ps
*τ_Li_*	low-frequency cutoff characteristic time	51.34 ps
*β_i_*	feedback strength	0.5~5 GHz
*τ_i_*	feedback delay time	2.5 ns
*γ_i_*	input gain	1

**Table 2 entropy-22-00231-t002:** Recognition results for the 8-bit, 16-bit, 32-bit optical packet headers when the SNR (signal-to-noise ratio) is 20 dB or 35 dB. NRMSE: normalized root mean square error.

	35 dB of SNR	8-bit Optical Packet Header Recognition		16-bit Optical Packet Header Recognition		32-bit Optical Packet Header Recognition	
20 dB of SNR							
Training NRMSE			0.04880		0.0879		0.0953
		0.0568		0.2003		0.2553	
Testing NRMSE			0.0870		0.1977		0.1650
		0.0954		0.3605		0.3725	
WER			0%		0%		0%
		0%		0%		0%	

**Table 3 entropy-22-00231-t003:** Digital speech recognition results for different SNR. WER: word error rate.

SNR of Input	Without Noise	30 dB	20 dB	10 dB
Training NRMSE	0.0509	0.0717	0.0729	0.1128
Testing NRMSE	0.1051	0.1136	0.1195	0.2485
WER	1.4%	1.6%	1.6%	14.6%

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
