# Peer review of "A Multiple-Input Multiple-Output Reservoir Computing System Subject to Optoelectronic Feedbacks and Mutual Coupling"

_entropy, 2020, doi:10.3390/e22020231_

Round 1
Reviewer 1 Report
In this paper, the authors numerically evaluate recognition performances in a multiple-input multiple-output reservoir computing system. The authors have discussed in a three-input system [19], and in this paper, extend it to the multiple-input system. The proposed system is different from the previous one slightly, but essentially they seem to be the same. The numerical evaluation shows the performances in the case of the four-input system. I cannot understand why this case should be evaluated, especially in terms of scientific meanings. If the contribution of the paper is a generalization of the number of the input/output, the authors should provide an evaluation as a function of the number. Also, I cannot find progress compared to the previous work; of course, the evaluation with noise may be a contribution, but it is a well-known fact that increasing noise depresses a performance. I think the results are trivial and easy to be expected, indicating that the submitted paper does not meet the acceptance level.
Author Response
Response to Reviewer 1:
1. In this paper, the authors numerically evaluate recognition performances in a multiple-input multiple-output reservoir computing system. The authors have discussed in a three-input system [19], and in this paper, extend it to the multiple-input system. The proposed system is different from the previous one slightly, but essentially they seem to be the same. The numerical evaluation shows the performances in the case of the four-input system. I cannot understand why this case should be evaluated, especially in terms of scientific meanings. If the contribution of the paper is a generalization of the number of the input/output, the authors should provide an evaluation as a function of the number.
Reply:
In our previous published literature [19] (now [17] X. Bao; Q. Zhao; H. Yin. Efficient optoelectronic reservoir computing with three-route input based on optical delay lines. Appl. Opt. 2019, 58, 4111-4117), we have done the numerical simulations of the three-input RC system. The results of the first edition of this paper refer to the simulations of a four-route RC system. The structures for the two proposed systems are also different. In Reference [19] (now [17]), a RC system utilizing the nonlinear characteristics of a semiconductor laser is proposed, while in this paper a RC system using the nonlinear characteristics of the Mach-Zehnder modulators is proposed. In other words, Reference [19] (now [17]) refers to an all-optical RC system, while this paper refers to an optoelectronic RC system.
At present, there are few papers published for the research of the multiple-input RC system. Multiple-input RC system is of great significance. For example, the switching nodes of optical networks usually have many input and output optical signals. Therefore, it is necessary to realize the multiple-route optical signal processing. In addition, the RC system has multiple inputs and outputs, which can also increase the processing speed and reduce the processing time. In the references listed in this manuscript, some parallel processing results using the photonic RC systems are also given. The two-route signal processing results are more. As far as we know, there are no reports on the four-route signal processing based on RC, let alone reports on the multiple input/output RC system.
In order to illustrate the feasibility of our proposed multiple-route RC system, we also add section 3.3. In this section, the simulation results of the 8-input/8-output optoelectronic RC system are provided to achieve the 8-route of 3-bit optical packet headers recognition. The simulation results of more input/output RC are not given here limited to the space of the manuscript. At the same time, the simulation also needs a long time. It is a good question proposed by the reviewer to evaluate the relationship between the route number and the performance of RC system. Next, we will do more in-depth research to try to provide a general rule.
2. Also, I cannot find progress compared to the previous work; of course, the evaluation with noise may be a contribution, but it is a well-known fact that increasing noise depresses a performance. I think the results are trivial and easy to be expected, indicating that the submitted paper does not meet the acceptance level.
Reply:
According to the reviewer's suggestion, we have revised the introduction section of the manuscript. Also we compared the scheme proposed in this manuscript with other references.
The influence of noise is seldom considered for the numerical simulations of the optoelectronic RC system in the current references. The influence of SNR on RC system is considered in reference [1]. This manuscript also analyzes the influence of SNR of the input signal on RC system.
[1] M. Soriano, S. Ortín, D. Brunner, L. Larger, C. Mirasso, I. Fischer, L. Pesquera. Optoelectronic reservoir computing: tackling noise-induced performance degradation. Optics Express, 2013, 21(1): 12-20.
Reviewer 2 Report
No comments to the authors.
Author Response
No comments to the authors.
Reviewer 3 Report
Dear Authors,
The paper is still little bit far from the definite version and high quality standards require. Please:
1) Check language and figures quality and presentation. Still broken figures.
2) Compare the method with another former method near in time.
3) What about other noise types ? How can they affect performance?
4) Limitations in the sense of implementation.
Regards
Author Response
Response to Reviewer 3:
The paper is still little bit far from the definite version and high quality standards require. Please:
1. Check language and figures quality and presentation. Still broken figures.
Reply: Thanks for the reviewer's comments, we carefully revised the manuscript. All the corrections have also been marked using the "Track Changes" function in Microsoft Word in the manuscript.
2. Compare the method with another former method near in time.
Reply:According to the reviewer's suggestion, we compare the RC system proposed in this manuscript with the parallel RC systems mentioned in other references near in time. We present the comparison in the second paragraph of the introduction section.
3. What about other noise types ? How can they affect performance?
Reply:Other noises in the RC structure mainly include the insertion loss noise of the Mach-Zehnder modulator, the quantum noise and the thermal noise of the photodetector. These noises are ignored in this manuscript. Only the noise of the input signal is considered. When considering the internal noise of the RC, the performance of the RC will be degraded.
The influence of noise is seldom considered for the numerical simulations of the optoelectronic RC system in the current references. The influence of SNR on RC system is considered in reference [1]. Our manuscript also analyzes the influence of SNR on the RC system.
[1] M. Soriano, S. Ortín, D. Brunner, L. Larger, C. Mirasso, I. Fischer, L. Pesquera. Optoelectronic reservoir computing: tackling noise-induced performance degradation. Optics Express, 2013, 21(1): 12-20.
4) Limitations in the sense of implementation.
Reply: At present, there are few papers reporting the RC system with multiple inputs. The multiple-input RC system is of great significance. For example, the switching nodes of the all-optical networks usually have multiple input optical signals. Hence, it is necessary to realize multiple-route optical signal simultaneous processing. In the references of this manuscript, some parallel processing results of the photonic RC are also listed, most of which are two-route input signals. As far as we know, the RC system with four-route signal processing has not been reported. Our scheme can realize the simultaneous processing of four-route signal, eight-route signal or more. Hence the proposed setup is more practical. With the development of optical devices and the optoelectronic integration technology, the scheme proposed here can be integrated.
Round 2
Reviewer 1 Report
The authors well revised the manuscript according to my comments. The paper can be published.
Reviewer 3 Report
Thank you for your comments. Please improve figures (cut ones) and revise again Language.